# Examining the Effect of MnS Particles on the Local Deformation Behavior of 8MnCrS4-4-13 Steel by In Situ Tensile Testing and Digital Image Correlation

Faisal Qayyum [1,*], Shao-Chen Tseng [2], Sergey Guk [1], Frank Hoffmann [1], Ching-Kong Chao [2] and Ulrich Prahl [1]

1   Institut für Metallformung, Technische Universität Bergakademie Freiberg, 09599 Freiberg, Germany; sergey.guk@imf.tu-freiberg.de (S.G.); frank.hoffmann@imf.tu-freiberg.de (F.H.); ulrich.prahl@imf.tu-freiberg.de (U.P.)
2   Department of Mechanical Engineering, National Taiwan University of Science and Technology, Taipei 106335, Taiwan; jason4071111@gmail.com (S.-C.T.); ckchao@mail.ntust.edu.tw (C.-K.C.)
*   Correspondence: faisal.qayyum@imf.tu-freiberg.de; Tel.: +49-373-139-4073

**Abstract:** In this study, the behavior of MnS particles in a steel matrix is investigated through in situ tensile testing and digital image correlation (DIC) analysis. The goal of this research is to understand the mechanical behavior of MnS inclusions based on their position in the steel matrix. To accomplish this, micro-dog bone-shaped samples are prepared, tensile tested, and analyzed. Macro-mechanical results reveal that the material yields at a stress of 350 MPa and has an ultimate tensile strength of 640 MPa, with a total elongation of 17%. For micro-mechanical analysis, scanning electron microscopy (SEM) images are taken at incremental strains and processed using DIC software to visualize the local strain evolution. The DIC analysis quantifiably demonstrates that the local strain is highest in the ferrite matrix, and while lowest in the pearlite matrix, the MnS particles and the interfaces between different materials experienced intermediate strains. The research provides new insights into the micro-mechanical deformation behavior of MnS particles in a steel matrix and has the potential to inform the optimization of the microstructure and properties of materials containing MnS inclusions.

**Keywords:** MnS inclusions; steel matrix; in situ testing; DIC analysis; mechanical behavior; microstructural analysis; strain evolution; material optimization

## 1. Introduction

Steel is a widely used material in construction and manufacturing due to its strength and versatility. However, pure steel has low ductility and strength, so it is often alloyed by adding interstitial and substitutional elements to improve its properties [1]. In steel alloys, various types of inclusion can be found that significantly influence the mechanical properties, microstructure, and overall performance of the material [2,3]. These inclusions are nonmetallic impurities that are either naturally occurring or introduced during the steelmaking process. A comprehensive understanding of the different types of inclusions is essential to optimize steel for specific applications. A common alloying element is sulfur, which is present in steel alloys due to the conventional steel production process and can also be added in small quantities to improve the machinability of block shapes that are near net [4]. However, sulfur can react with nonmetallic inclusions in steel to form sulfides [5]. MnS inclusions are known to promote machinability and act as chip breakers, but can also negatively affect steel toughness, fatigue resistance, and hot ductility [6]. Past research on sulfide inclusions in steel has focused primarily on the effect of sulfur percentage, manufacturing process, and heat treatment on the amount, shape, size, and properties of sulfides [7–11]. For example, studies have shown that increasing the sulfur content in steel can lead to the formation of larger andmore elongated sulfides, which can negatively

impact the machinability and mechanical properties of the material [12]. Similarly, certain manufacturing processes and heat treatments can alter the morphology and distribution of sulfides in the steel matrix, which can affect performance in service [13].

Another common type of inclusion is the inclusion of alumina ($Al_2O_3$), which comes from the deoxidation process involving aluminum [14]. Alumina inclusions can affect steel cleanliness and their size and distribution can influence steel mechanical properties [15]. Furthermore, calcium aluminates ($CaO\text{-}Al_2O_3$) are complex inclusions that can form when calcium is added to modify the size and shape of alumina inclusions, ultimately improving castability and cleanliness [16]. Silicate inclusions, which consist of various combinations of silicon, oxygen, and other elements such as calcium or magnesium, can also be found in steel, affecting its properties and performance [16].

In addition to these primary types of inclusions, steel can contain a variety of other complex inclusions, such as titanium nitrides (TiN), titanium carbonitrides (Ti(C,N)), and titanium oxides (TiOx) [15]. These inclusions arise from the interaction of titanium with nitrogen, carbon, and oxygen, respectively. They can serve as grain refiners, improving steel mechanical properties, but can also create local stress concentrations, potentially compromising steel integrity [17].

Understanding the effects and behavior of inclusions in steel during deformation and damage is crucial to optimizing material performance. Various tests and studies have been conducted to analyze these inclusions and their impact on steel properties. For example, techniques such as scanning electron microscopy (SEM) and transmission electron microscopy (TEM) allow the characterization of inclusions in terms of size, shape, and chemical composition [14,18]. Coupling these microscopy techniques with energy-dispersive X-ray spectroscopy (EDS) provides detailed information on the elemental distribution within inclusions [19–21]. Furthermore, electron backscatter diffraction (EBSD) can be used to investigate the relationship between inclusions and the surrounding microstructure, including grain orientation and misorientation [22]. This information is valuable for understanding the impact of inclusions on the nucleation and propagation of cracks in steel.

In addition to these experimental approaches, computational methods, such as crystal plasticity finite element modeling (CPFEM) [23], have been used to study the behavior of inclusions during deformation and damage at microstructural scales. These computational techniques can help predict the stress–strain response, dislocation density, and crack initiation and propagation behavior around inclusions, providing critical information on the performance under various loading conditions.

Although extensive research on the effect of sulfur percentage, manufacturing process, and heat treatment on the properties of sulfides has been conducted, there is still a lack of understanding of the local deformation behavior of these inclusions and their interactions with the matrix. Previous research has focused on steel effect of sulfides on the machinability and mechanical properties [9], but has not thoroughly investigated the particle–matrix interface. To address this research gap, this study aims to investigate the local deformation behavior of sulfide inclusions in a steel matrix using miniature samples that are manufactured with appropriate heat treatment and machining. The samples are deformed under quasi-static tensile conditions in an electron microscope, and high-resolution images are recorded and post-processed using digital image correlation (DIC) software to analyze the local engineering strain.

## 2. Materials and Methods

### 2.1. Material

The hot rolled 8MnCrS4-4-13 carbon steel utilized in this study has a chemical composition (wt%) of 0.08 C, 0.04 Si, 1.07 Mn, 1.07 Cr, 0.01 P and 0.13 S. This particular steel is chosen due to its capacity to form a substantial amount of manganese sulfide (MnS) inclusions. The sulfur content in this material is significantly elevated compared to that of conventional sulfur-alloyed case-hardened steels, which can be attributed to the addition of

iron sulfide (FeS). When FeS is added, sulfur preferentially reacts with manganese, forming MnS inclusions that are thermodynamically stable in the molten state.

To process the material, a normalizing hot rolling procedure is performed, with a final rolling temperature of approximately 900 °C. This treatment is followed by air cooling, resulting in a 16 mm steel wire. Subsequently, spheroidization annealing is carried out, consisting of a heat treatment for 15 h at 720 °C. This step was aimed at achieving a spheroidized microstructure with large grains, which is particularly advantageous for this study. A metallographic examination of the steel samples is performed after etching with a 3% nital solution, which selectively attacks the ferrite grain boundaries and pearlite grains. The etching process revealed the presence of ferrite and pearlite in the microstructure, providing insight into the microstructural constituents.

### 2.2. Methods

A range of advanced experimental techniques and tools are employed to thoroughly investigate the microstructural deformation behavior of steel alloys containing MnS particles. Microtensile samples are prepared using appropriate heat treatment and machining procedures to ensure that the microstructure of the material is representative of the bulk material. The size and dimensions of these samples are shown in Figure 1. In situ tensile testing is then used to study the deformation behavior of the samples under realistic loading conditions.

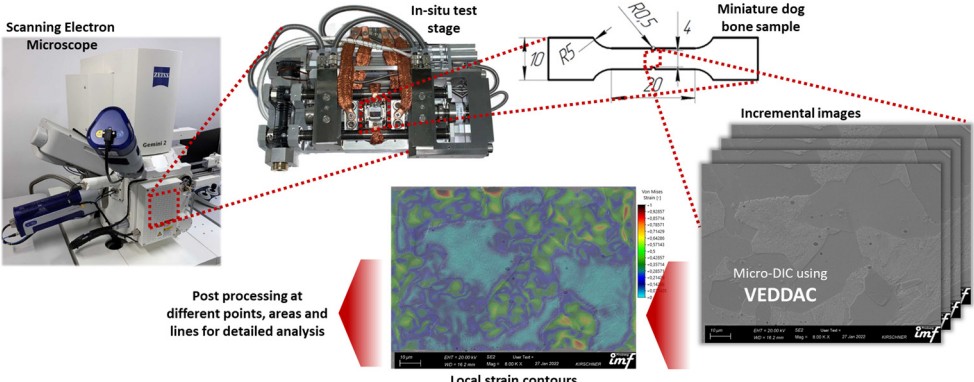

**Figure 1.** A schematic figure to show the workflow, sample geometry, and steps carried out during this study.

In this study, a Zeiss Gemini scanning electron microscope (SEM) was used to capture high-resolution images of the steel microstructure at various strain levels. The SEM is equipped with secondary electron (SE), backscattered electron (BSE), electron backscatter diffraction (EBSD), and energy-dispersive X-ray spectroscopy (EDS) detectors, enabling comprehensive characterization of the microstructure and elemental distribution. Steel samples are mounted on a Kammrath and Weiss tensile/compression test stage and subjected to tensile testing at a quasi-static strain speed of $1 \times 10^{-3}$ s$^{-1}$ with intermediate interruptions to capture images. The tensile/compression module used in this study uses a load cell with a maximum load capacity of 10 kN. The strain gauge-based load cell has a precision class of 0.3% and offers a combined error (linearity, hysteresis) of 0.3% on the full scale.

High-resolution images are captured at regular intervals throughout the deformation process. These images are subsequently processed using the digital image correlation (DIC) software VEDDAC, which facilitated visualization of the local strain evolution in distinct regions of the material. To minimize the effects of carbon build-up on the sample surface and prevent ratcheting and increased plasticity, certain precautions are taken during the testing process. Specifically, the testing is temporarily stopped for short periods, and the frequency of image acquisition is reduced in regions exhibiting higher plasticity. It is also important to recognize that the microstructure in the third dimension can significantly

influence the surface behavior of a material. In this study, the analysis focused solely on the surface of the sample, which could exhibit a different behavior from that of the bulk material [24–26]. Although previous research has used computed tomography (CT) to investigate the 3D microstructure of materials [27,28], we found that using a scanning electron microscope (SEM) in our study resulted in considerably higher resolution images. Despite the limitation of analyzing only the sample surface, high-resolution SEM imaging offered a complete understanding of the microstructure, producing more accurate and insightful results than would have been achievable with CT.

After the tensile test, the force–displacement data is processed to generate the engineering stress–strain curve, as shown in Figure 2. Subsequently, essential mechanical properties, such as elastic modulus, yield point of 350 MPa, maximum tensile strength (UTS) of 727 MPa, and total elongation of 17.6%, are calculated for the material under investigation are calculated. Figure 2 shows an elastic modulus (E) of 727 MPa for the tested steel, which is significantly lower than the typical ~210 GPa observed for steels. This anomaly is due to the use of an in situ tensile test stage, which lacks a strain gauge. Therefore, the measured initial deformation of the sample also incorporates the mechanical stiffness of the entire Kammrath and Weiss measuring system. This includes moving parts, load cell beams, and slight bending of holding grips, which collectively reduce the measured elastic stiffness. Each kink in the shown stress–strain curve is due to a break in deformation for recording images.

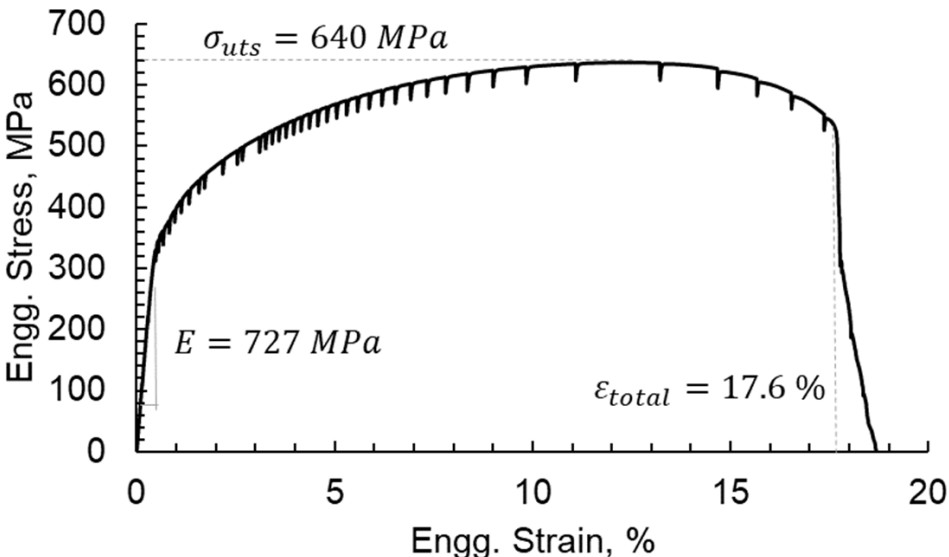

**Figure 2.** Engineering stress–strain curve of the in situ tensile test showing the global response of the steel alloy sample. The elastic modulus, the yield point, the ultimate tensile strength (UTS), and the total elongation of the material are calculated from the curve. Each kink in the curve represents a stopping point for the test to record an SEM image.

Figure 3 illustrates the evolution of surface distortion and changes in grayscale values of the sample at various strains chosen during in situ tensile testing. Surface distortions can adversely affect the accuracy of strain measurements obtained by digital image correlation (DIC) analysis. To address this issue, appropriate image preprocessing techniques, such as image smoothing or intensity correction, were employed.

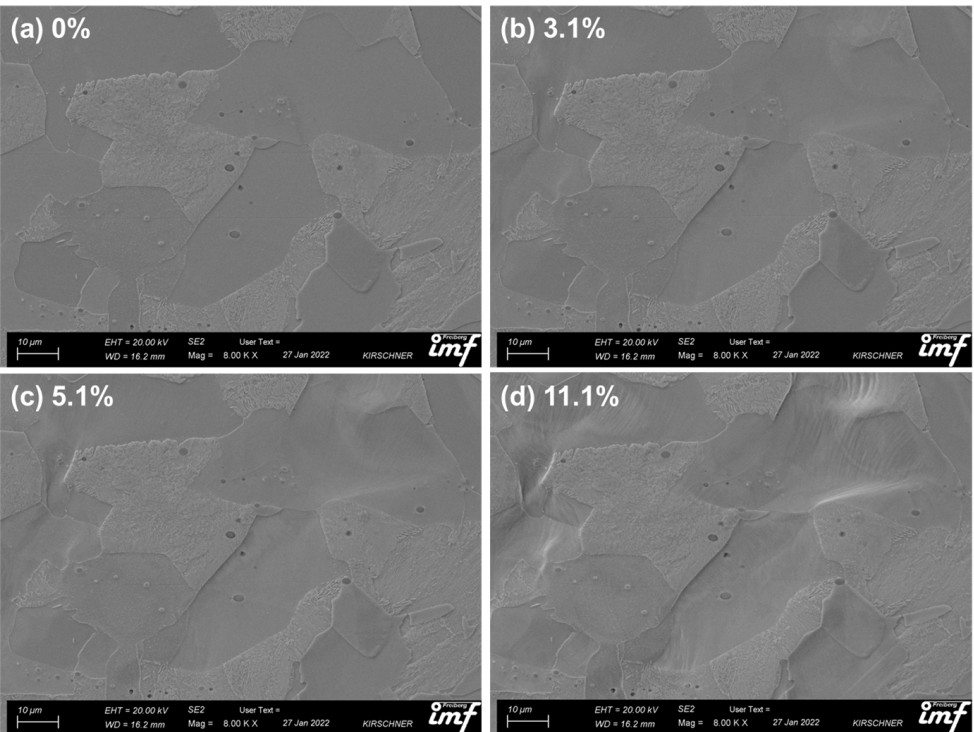

**Figure 3.** Few of the many SEM micrographs of the steel alloy sample with deformation of 0%, 3.1%, 5.1% and 11.1% deformation during in situ tensile testing. Images from (**a–d**) demonstrate the evolution of surface distortion and changes in the grayscale value.

### 2.3. Digital Image Correlation

The post-processing of the SEM micrographs obtained from the interrupted in situ tensile test is conducted using the commercial DIC tool VEDDAC. The image-based DIC technique functions by defining an initial major and minor grid. The main grid divides the image into fixed sections, while the minor grid assigns a unique grayscale value to a small section of the image. This unique grayscale value is tracked across each input image to calculate a velocity vector for that section of the image. The calculated velocity vectors can then be post-processed to obtain local strain fields.

In this study, a total of 21 SEM micrographs are utilized with a dynamic reference. The complete image data set is provided in Appendix A for further reference and use. Each micrograph has a resolution of 2048 × 1536 pixels. A grid pitch of 15 × 15 pixels is used to divide the image into equal sections. A measuring field of 80 square pixels is defined around each grid patch, with a reference field of 20 square pixels acting as a grayscale fingerprint. The reference field is tracked throughout the images to track the motion of every material point to obtain the velocity field. To clean the processed velocity field, a Gaussian radius of 4 pixels is defined, and the outlier detection sensitivity is set to 20. A strain radius of 4 pixels is defined to calculate the strain field from the velocity field.

For post-processing of the results to analyze the effect of MnS particles on the local deformation behavior, several areas of interest are defined in the initial micrograph. The areas of interest are illustrated in Figure 4, with inclusions 1, 2, 5, and 8 (shown in blue) present entirely within the pearlite grains, inclusions 4 and 6 (shown in yellow) present entirely within the ferrite grains, and inclusions 3 and 7 (shown in red) present on the grain boundaries of ferrite and pearlite.

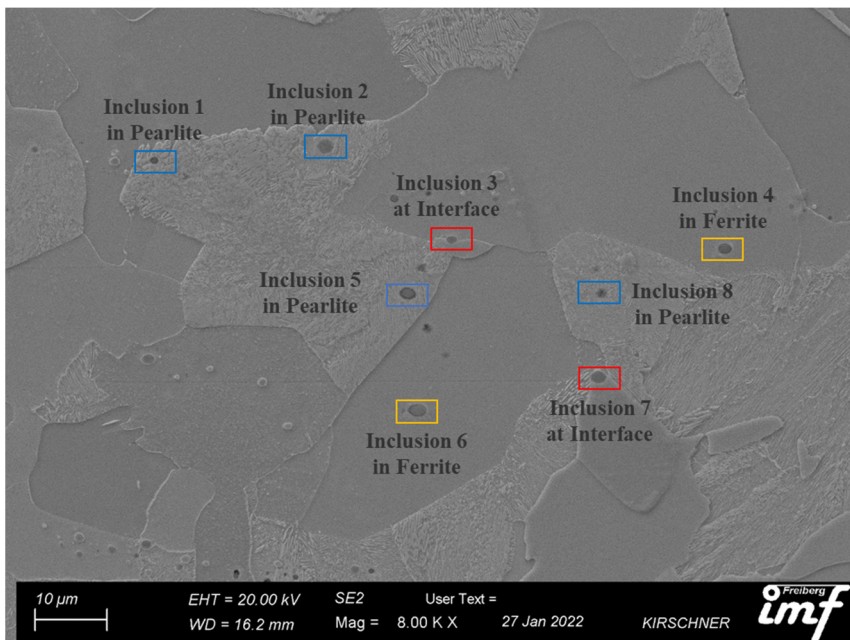

**Figure 4.** SEM micrograph of the steel alloy sample with identified areas of interest, represented by inclusions 1, 2, 5, and 8 (shown in blue) present entirely within the pearlite grains, inclusions 4 and 6 (shown in yellow) present entirely within the ferrite grains, and inclusions 3 and 7 (shown in red) present on the grain boundaries of ferrite and pearlite.

## 3. Results and Discussion

The objective of this study is to investigate the microstructural deformation behavior of steel alloys containing MnS particles using advanced experimental techniques and tools. To achieve this, miniature dog bone samples are prepared and subjected to in situ tensile tests, while SEM images are captured at various strains. The results, as illustrated in Figure 2, showed that the material exhibited yield at a stress of 350 MPa and an ultimate tensile strength of 640 MPa, with a total elongation of 17%. To visualize the local strain evolution in different areas of interest, including MnS particles within pearlite and ferrite grains and on the interface of pearlite and ferrite grains, the SEM images are processed using digital image correlation (DIC) software. The results of the study are analyzed and presented in two ways. First, the results are overlaid on the micrographs in a color contour to qualitatively present the local strain field and how it is affected by the heterogeneous microstructure. Secondly, the local strain distributions for the areas of interest are plotted alongside the evolving global strain to present the evolution quantitatively.

To provide a better understanding of our discussion, it is important to define some of the terms about material properties. Within the context of this study, the terms 'soft Ferrite' and 'hard Pearlite' are used to describe different phases of steel. Ferrite is a soft, ductile phase of iron. It is body-centered cubic (bcc) in its crystal structure and is magnetic. Its softness comes from its crystal structure, which allows for a significant degree of plastic deformation before failure. Pearlite, on the other hand, is a two-phased, lamellar (or layered) structure composed of alternating layers of ferrite and cementite (an iron carbide), which occurs in steel when it is cooled slowly. Pearlite is stronger and harder than ferrite as a result of the presence of a hard cementite phase in its structure. When we refer to 'softer regions', we generally indicating areas dominated by the ferrite phase. The 'stronger pearlite grains' and 'hard pearlite phase' refer to areas where the pearlite phase is prevalent.

### 3.1. Local Strain Contour

The results presented in Figure 5 demonstrate the local equivalent Mises strain distribution overlaid on the micrographs using a color contour, which qualitatively shows the local strain field and how it is affected by the heterogeneous microstructure. The

initial image with zero strain (Figure 5a) shows ferrite grains marked with orange dots and pearlite grains marked with yellow dots.

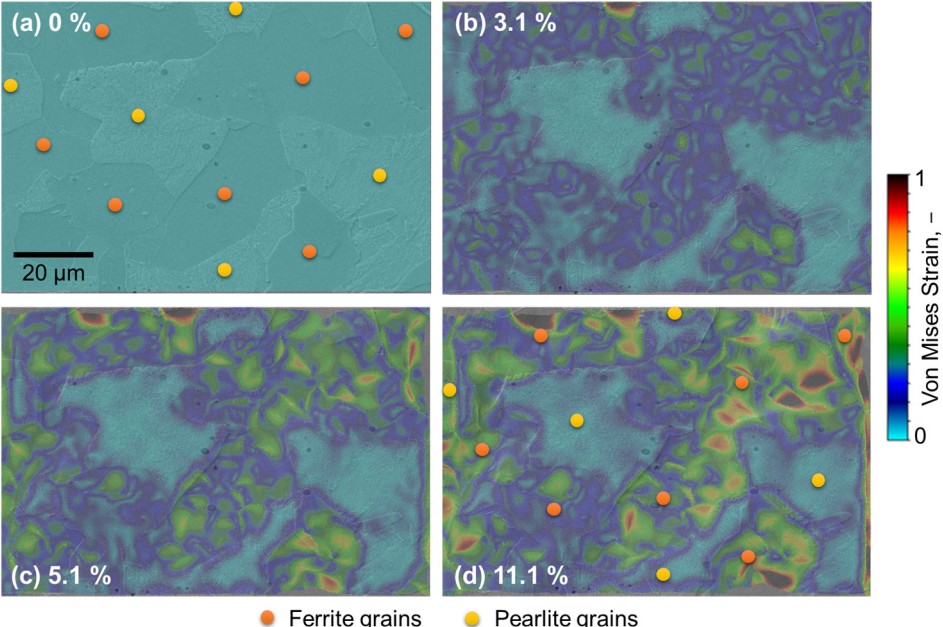

**Figure 5.** The local equivalent Mises strain distribution overlaid on micrographs of multiphase steel under progressive global strain. The initial image with zero strain is shown in panel (**a**), where the ferrite grains are marked with orange dots, and the pearlite grains are marked with yellow dots. The plots (**b**–**d**) show the same microstructure under a global strain of 3.1, 5.1, and 11.1, respectively.

The results show that as the global strain progresses from 3.1 to 5.1 to 11.1 in Figure 5b–d, respectively, the local equivalent Mises strain in ferrite grains evolves to a much higher level of approximately 100% in some regions, while the strain in pearlite grains remains close to zero. This observation highlights how deformation affects the different phases of the steel in distinct ways, which is critical to understanding the mechanical behavior of multiphase steels.

Furthermore, the results indicate that the inclusions present in the ferrite grains are surrounded by lower strain regions because they work to hold the ferrite grain together, whereas in pearlite grains, the MnS particles are surrounded by slightly higher strain, as they ensure softer regions within the stronger pearlite grains. This observation highlights the role of inclusions, especially MnS, during deformation and how they contribute to the overall mechanical behavior of the material, which is consistent with previously published observations [29,30].

### 3.2. Local Strain in the Matrix and Individual Particles

To investigate the microstructural deformation behavior of steel alloys containing MnS particles, the localities in the initial SEM images are divided into three different categories, which are shown in Figure 6. These categories included: MnS particles present within pearlite grains, represented by solid blue squares; MnS particles present within ferrite grains, represented by solid yellow squares; and MnS particles present on the interface of pearlite and ferrite grains, represented by solid red squares. To gain a comprehensive understanding of the deformation behavior of these particles, we track the local deformation evolution in each of these categories using two different techniques. The first technique involved monitoring the strain throughout the area, while the second technique involved monitoring the strain in specific, localized areas, represented by dotted (blue, yellow, and red) rectangles. These two techniques are chosen to provide data from both a global material point of view and a more localized perspective.

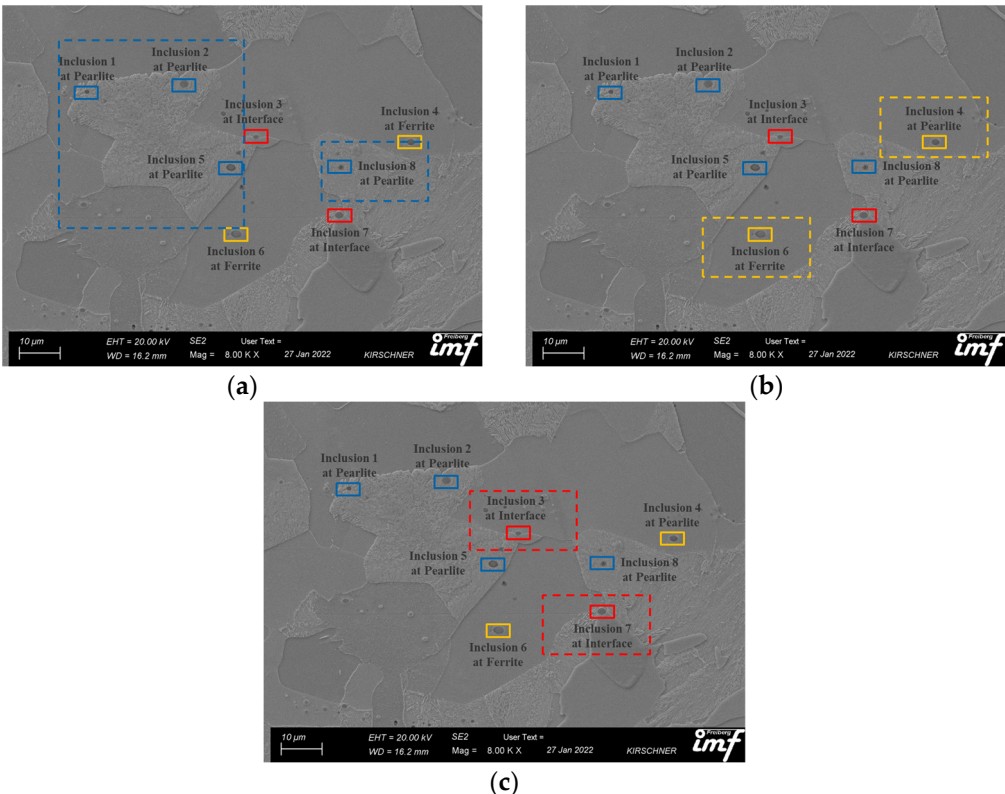

**Figure 6.** The local von Mises strain distribution in different regions of interest is represented by (**a**) blue zones representing MnS particles within pearlite grains, (**b**) yellow zones representing MnS particles within ferrite grains, and (**c**) red zones representing MnS particles on the interface of pearlite and ferrite grains.

The results of the study on the local deformation behavior of MnS particles are presented in Figure 7. In Figure 7a, the strain in individual MnS particles is plotted against the global strain (strain throughout the entire SEM image). Inclusions 7 and 6 exhibit the highest strain, around 0.5, which is comparable to the general strain in the matrix. Inclusions 3 and 4 exhibit moderate strain, between 0.2 and 0.3, which increases initially and then decreases toward the end of the test. This decrease in strain may be caused by the activation of the local compressive strain once the alternate slip systems in the grain are activated.

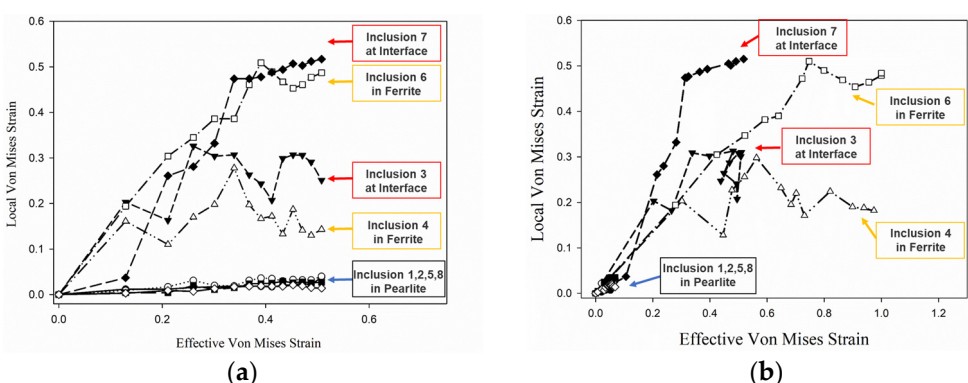

**Figure 7.** Local deformation behavior of MnS particles. (**a**) Comparison of strain in individual MnS particles with global strain, and (**b**) comparison of strain in individual MnS particles with local strain in different regions of interest, as defined in Figure 6.

In Figure 7b, the strain in individual MnS particles is plotted against the local strain (strain within the specific areas defined by dotted lines in Figure 6). It is observed that, depending on the matrix material, the evolution of local strain is different. The local strain reaches up to 1.0 in the ferrite matrix, is almost zero in the hard pearlite phase, and is around 0.5 (the mean of both cases) in the interface areas. These results demonstrate that MnS particles embedded in the hard pearlite phase undergo almost negligible strain, while those in the ferrite matrix undergo high strain, and those on the interface experience a strain in between the extremes mentioned above. These trends are intuitive; however, a more detailed study of MnS particles on the interface of both phases is carried out in the next part of this work to gain further insight into their deformation behavior.

### 3.3. Local Strain along the Particle/Matrix Interface

The local deformation behavior of MnS particles at the interface between different matrix materials is a key area of interest in this work. To further understand the unique behavior of these particles, we have analyzed the local strain evolution at the interface of a specific inclusion (inclusion 7). The deformation behavior at five key points along the interface, including the ferrite matrix, the ferrite–MnS interface, the MnS particle itself, the MnS–pearlite interface, and the pearlite matrix. Figure 8 provides a visual representation of the different points and regions studied, which can help to understand the results of this analysis.

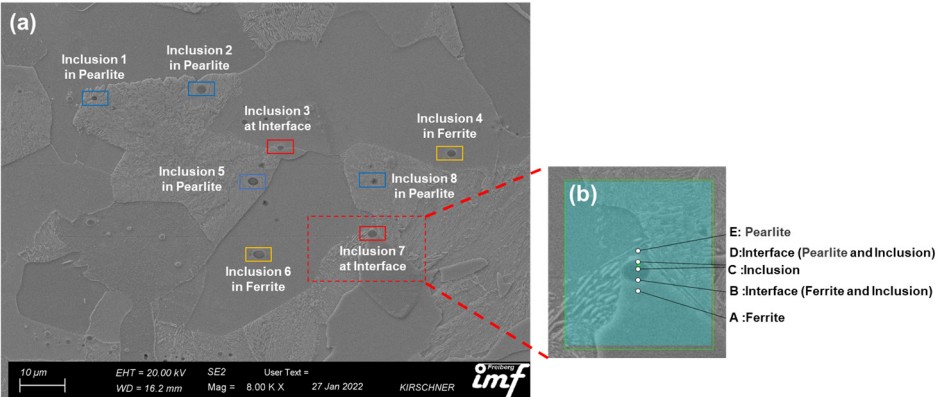

**Figure 8.** Analysis of local deformation behavior on the MnS particle interface of inclusion 7. (**a**) SEM micrograph of the inclusion studied, and (**b**) representation of the measurement points labeled ferrite, ferrite–MnS interface, MnS particle, MnS-pearlite interface and pearlite.

In Figure 9, the local strain evolution along the five different measurement points (ferrite, ferrite–MnS interface, MnS, MnS–pearlite interface, pearlite) is presented with respect to the local strain evolution in the local area (shown by the dotted red rectangle). It is evident from the data that there is almost no strain present in the pearlite matrix, as reflected by the extremely low strain on the inclusion–pearlite interface, which is less than 0.2%. On the other hand, a significant amount of strain is observed in the ferrite matrix, with a strain of more than 2.5%, resulting in a correspondingly large strain at the inclusion–ferrite interface, with a strain of more than 0.6%. This large variation in the local strain around the MnS particle leads to a high contrast of local shear strain and is believed to be a critical factor in the formation of voids and growth phenomena. This is supported by previous research [29–33], in which scientists reported that the high shear strain pockets around the inclusion particles are responsible for the high local triaxiality and eventual initiation of local damage.

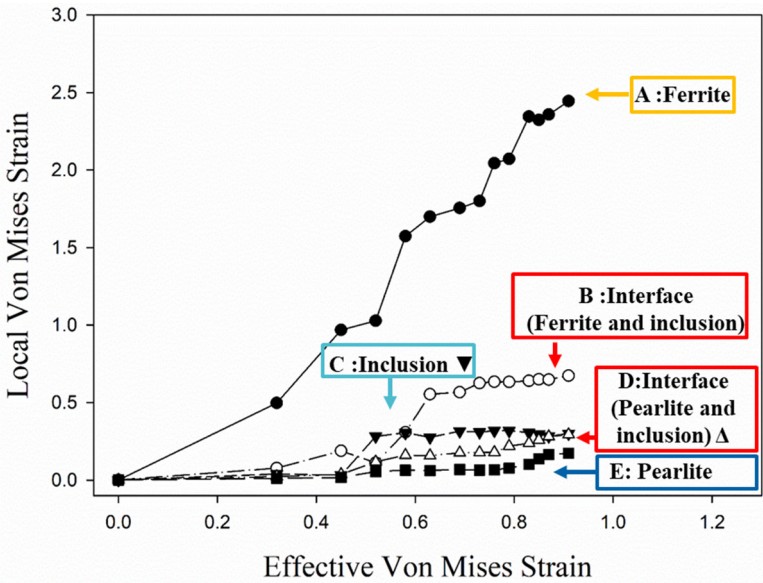

**Figure 9.** Evolution of different points of measurement along the MnS–matrix interface in inclusion 7.

Our results showed that the microstructural deformation behavior of the material is greatly influenced by the position and morphology of the MnS particles. The highest locally normalized strain is observed in inclusions 4 and 6, both of which are entirely embedded within the ferrite grains. Inclusions 1, 2, 5, and 8, located entirely within the pearlite grain, showed the lowest strain. The medium strain is found in inclusions 3 and 7, which are on the grain boundaries of the ferrite and pearlite grains. The study demonstrated that the evolution of local strain varies depending on the matrix material. Local strain reached a maximum of 1.0 in the ferrite matrix, is nearly zero in the hard pearlite phase, and approximately 0.5 in the interface areas. This indicates that MnS particles embedded in the hard pearlite phase experience almost negligible strain, whereas those in the ferrite matrix undergo a very high strain, and particles on the interface experience a strain in between these extremes. These results offer a deeper insight into the deformation behavior of MnS particles in steel alloys and could have implications for the development of new materials.

It should be noted that the present study has some limitations. The imaging resolution cannot go beyond 2D, and the free surface of the sample may behave differently from the bulk material. Furthermore, this study is limited to a specific alloy and the results may not be generalizable to other alloys or materials. However, the results and methodology used in this work could be used as a basis for future studies with further advanced techniques and tools to study the microstructural deformation behavior in more detail.

## 4. Conclusions

The findings of this study have significant implications for the development of new materials and for understanding the deformation behavior of materials with inclusions. We found that the inclusion of MnS particles leads to a different deformation behavior, depending on their position and morphology. The inclusions located within the pearlite grains showed the highest resistance to deformation, while the inclusions located within the ferrite grains showed the highest susceptibility to deformation. The inclusions located on the grain boundaries showed an intermediate level of deformation behavior. These findings provide new insights into the local deformation behavior of steel alloys around MnS particles, particularly with respect to the impact of their location and the surrounding matrix phase. Our study highlights the nuanced interaction between MnS inclusions and their host matrix, which could prove beneficial for further research in the field. Although our findings do not directly suggest strategies to manipulate the distribution of these inclusions, they do underscore the importance of considering their effect on microstructural

behavior, which is relevant for industries such as automotive, aerospace, and energy, where steel alloys are widely used. Some specific conclusions from this study are as follows:

1. The yield of the material occurs at 350 MPa with a UTS of around 640 MPa and a total elongation of 17%;
2. The highest strain on the particles is observed in the inclusions present in the ferrite matrix (approximately 2.5%);
3. MnS particles embedded in the hard pearlite phase undergo almost negligible strain (approximately 0.1%);
4. The particles on the interface of ferrite and pearlite grains experience strain in the middle of the previously mentioned extremes, approximately 0.5 (mean of both cases);
5. The high shear–stress pockets around inclusion particles might be responsible for the high local triaxiality, which drives the local void nucleation and growth phenomenon.

The methodology and techniques used in this study, such as in situ tensile testing and DIC, can be applied by other scientists in their research to study the deformation behavior of other materials or investigate other mechanical properties.

**Author Contributions:** Conceptualization, S.-C.T. and F.Q.; methodology, S.-C.T.; software, F.H.; validation, S.-C.T., F.H. and F.Q.; formal analysis, S.-C.T. and F.H.; investigation, S.-C.T.; resources, F.Q.; data curation, F.Q.; writing—original draft preparation, S.-C.T. and F.H.; writing—review and editing, S.-C.T., F.H., C.-K.C. and S.G.; visualization, S.-C.T., F.Q., C.-K.C. and S.G.; supervision, C.-K.C., S.G. and U.P.; project administration, U.P. All authors have read and agreed to the published version of the manuscript.

**Funding:** The stay of Tseng at IMF was sponsored by the DAAD/MOST scholarship (Sandwich program) for doctoral candidates from Taiwan (No. 110-2927-I-011-505). The stay of Qayyum at the IMF was partially sponsored by the DAAD Faculty Development for Candidates (Balochistan), 2016 (57245990) HRDI-UESTP/UET funding scheme in cooperation with the Higher Education Commission of Pakistan (HEC).

**Data Availability Statement:** Data, software, and scripts from this study are not publicly available, but can be shared with interested researchers and groups upon request. Please contact Faisal Qayyum at www.faisalqayyum.com/contact (accessed on 14 July 2023) for such a request.

**Acknowledgments:** The authors acknowledge the DAAD/MOST scholarship and DAAD Faculty Development Funding Scheme for Candidates HRDI-UESTP/UET funding scheme in cooperation with the Higher Education Commission of Pakistan (HEC). We acknowledge the contributions of Markus Kirchner for helping with the SEM data collection.

**Conflicts of Interest:** The authors declare that they have no conflict of interest.

**Appendix A**

All the SEM images are provided below for further details and consideration for the readers.

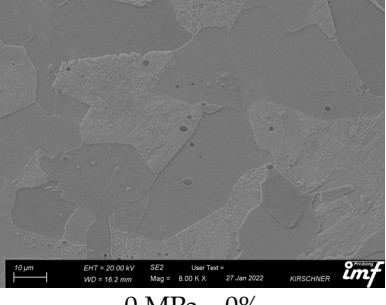 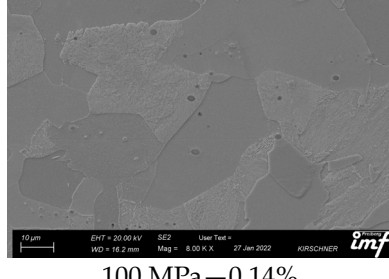 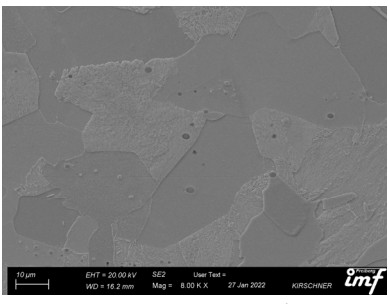

| | | |
|:---:|:---:|:---:|
| 0 MPa—0% | 100 MPa—0.14% | 300 MPa—0.47% |

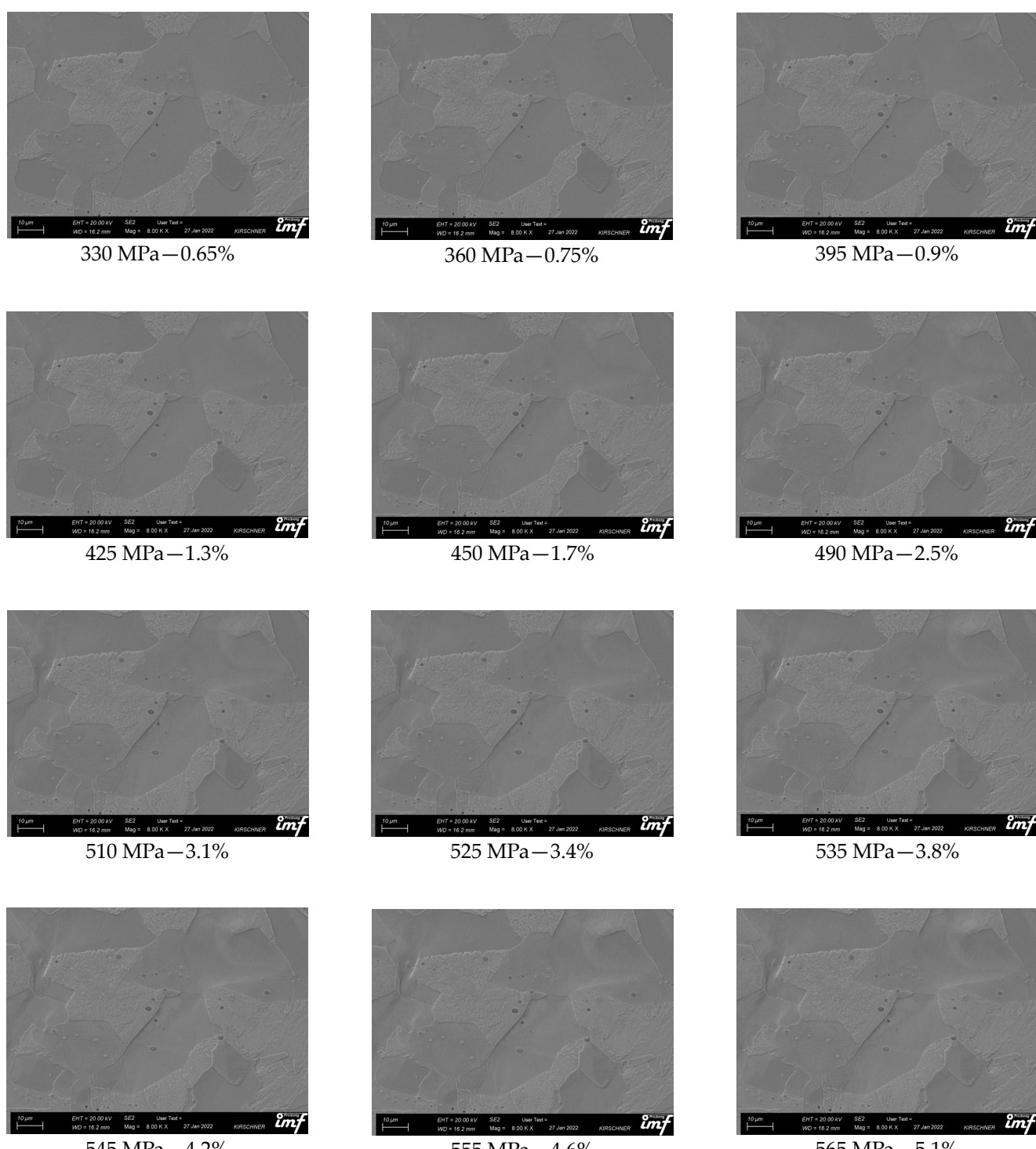

330 MPa—0.65%  360 MPa—0.75%  395 MPa—0.9%

425 MPa—1.3%  450 MPa—1.7%  490 MPa—2.5%

510 MPa—3.1%  525 MPa—3.4%  535 MPa—3.8%

545 MPa—4.2%  555 MPa—4.6%  565 MPa—5.1%

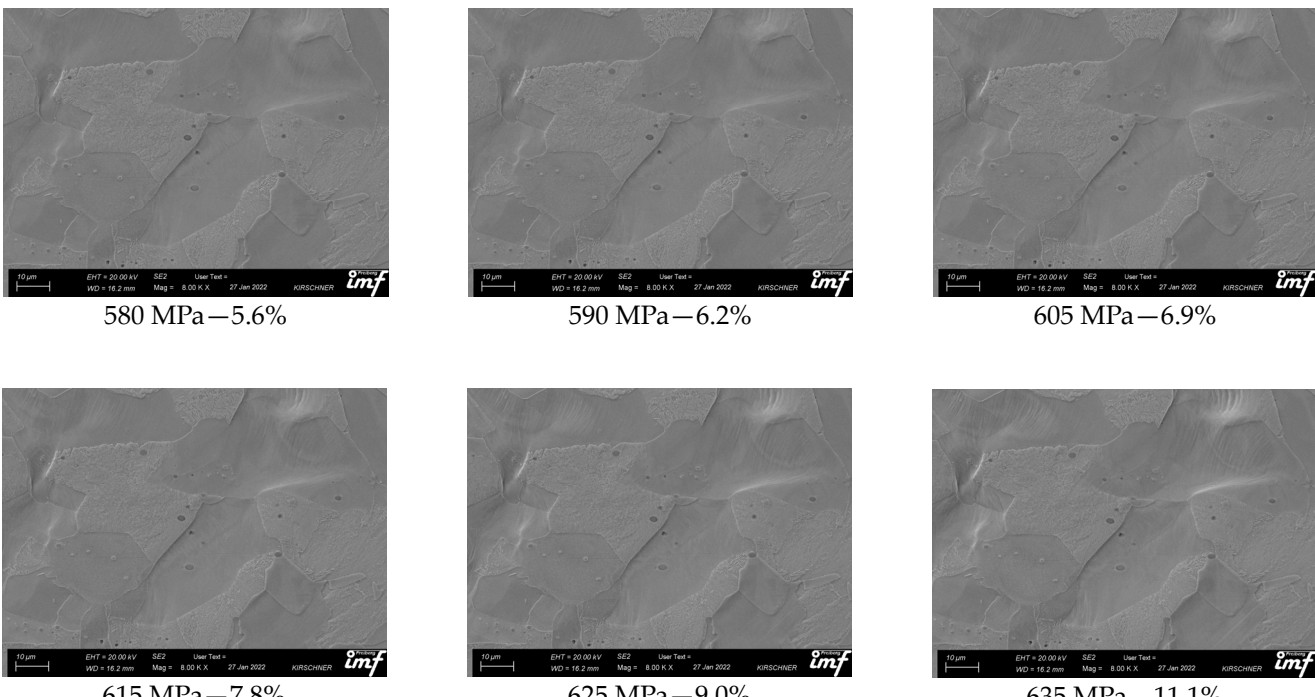

580 MPa—5.6%        590 MPa—6.2%        605 MPa—6.9%

615 MPa—7.8%        625 MPa—9.0%        635 MPa—11.1%

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
