# Peer review of "Examining the Effect of MnS Particles on the Local Deformation Behavior of 8MnCrS4-4-13 Steel by In Situ Tensile Testing and Digital Image Correlation"

_jcs, doi:10.3390/jcs7070294_

Round 1

Reviewer 1 Report

The deformation behavior of MnS particles with strain was investigated. There are several aspects that should be addressed to improve the quality of the work.

1.     What does “miniature samples” mean? Normally, there are two kinds of tensile samples, i.e standard sample and micro-tensile sample with very small size.

2.     Fig.1, elastic modules of the tested steel is E=727 MPa, which is much lower than the normal value of ~210GPa. The tensile results are questionable.

3.     Lines 176-177, “Figure 2 illustrates the evolution of surface distortion and changes in grayscale values of the sample at various strains during in-situ tensile testing.”

It is difficult to observe information on the evolution of surface distortion and changes in grayscale values of the sample at various strains.

4.     It is not necessary to capitalize the initial letter of ferrite and pearlite.

5.     Lines 235-237, “To visualize the local strain evolution in different areas of interest, including MnS particles within Pearlite grains, MnS particles within Ferrite grains, and MnS particles on the interface of Pearlite and Ferrite grains, …”

It is better to be replaced by “To visualize the local strain evolution in different areas of interest, including MnS particles within pearlite and ferrite grains, and on the interface of pearlite and ferrite grains, …”

6.     Lines 337-340, “Specifically, the study found that the highest strain was observed in inclusions 7 and 6, which were entirely embedded within the Ferrite grains. The lowest strain was observed in inclusions 3 and 4, which were located on the grain boundaries of Ferrite and Pearlite grains.”

But inclusion 7 is not entirely embedded within the ferrite grains, and inclusion 4 in not on the grain boundaries of ferrite and pearlite grains (it is in ferrite!)

7.     Lines 362-367, “the inclusion of MnS particles could be used to improve the mechanical properties of these materials. Additionally, our results may have important implications for the use of MnS particles in a wide range of industrial applications, such as in the automotive, aerospace, and energy industries”

This statement is not suitable because of following reasons: (1) the relationship between properties and inclusions is not studied in the study; (2) no ideas on how to control distribution of inclusions are given. In fact, MnS inclusions have a negative effect on mechanical properties of steels. The control of inclusion distribution is almost impossible in production (for example, to control inclusion to be in pearlite region).

8.     There is no discussion on inclusion morphology, so the title should be revised.

Author Response

Please see the attached PDF file with detailed response to each comment. 

Reviewer 2 Report

This article proposes a characterization of the microstructural behavior of MnS particles in a steel matrix through in-situ tensile testing and digital image correlation (DIC). The goal is to understand the mechanical behavior of MnS inclusions and (their interactions with the matrix) based on their position, size and distribution in the steel matrix.

The manuscript is well written and it is an interesting work for the community concerning by the mechanical characterization. However, some aspects require clarification:

The chemical composition (wt %), L. 105: the number of significant figures should be consistent for the same physical quantities.

L. 119: Ferrite ->ferrite, Pearlite -> pearlite

L. 139: quasi-static speed? Value?

Capacity of the load cell and the associated resolution?

“softer regions, stronger Pearlite grains, hard Pearlite phase…”: it is not evident for a non-specialist to understand these terms. A clear definition of these terms at the beginning of the explanations would be useful the readers.

Figure 6 an others: is it possible to obtain a value or at least an indication of the errors associated with local strain measurements?

Author Response

(The authors gave the same response as above.)

Reviewer 3 Report

The paper is well written and clearly analyzes interesting data on the mechanical behavior of individual MnS inclusions in domains of ferrite and pearlite. However, to publish the work, some minor corrections are needed as follows.

1.      The first paragraph of Section 3 duplicates wording of Section 2. It is reasonable here to shorten the text by merely mentioning Section 2.

2.      It is unclear in comments to Fig. 3 which COMPONENT or scalar MEASURE of strain is mapped by DIC. Although “Von Mises” appears at a color bar of the PICTURE, it would be better to directly say in the TEXT that the equivalent strain is considered.  

3.       Rather confusing statement in the last paragraph before Conclusions: 

According to the authors, MnS particles demonstrate whether the highest “RESISTANCE” or the highest “SUCCEPTIBILITY” to strain when situated in pearlite or ferrite, respectively. Meanwhile, as indicated by the obtained results, so dissimilar properties should be ascribed to the pearlite or ferrite THEMSELVES rather than respective (similar) inclusions. The latter just follow lower or higher deformations of their surroundings and, naturally, produce stronger strain singularities when interacting with a stronger deformed phase of steel.     

Author Response

(The authors gave the same response as above.)

Round 2

Reviewer 1 Report

Authors revised manuscript according to comments, added more information and clarified some points. Revised manuscript is more reasonable and readable.

Reviewer 2 Report

the authors' comments and explanations answer my questions

Reviewer 3 Report

Now, as my remarks were properly adressed, the paper is recommended to publish.